# Underwater Target Recognition Based on Improved YOLOv4 Neural Network

**Lingyu Chen, Meicheng Zheng, Shunqiang Duan, Weilin Luo *** and **Ligang Yao**

College of Mechanical Engineering and Automation, Fuzhou University, Fuzhou 350108, China;
feixiacly@gmail.com (L.C.); zmcheng@163.com (M.Z.); dsq290187596@163.com (S.D.); ylgyao@fzu.edu.cn (L.Y.)
* Correspondence: wlluo@fzu.edu.cn

**Abstract:** The YOLOv4 neural network is employed for underwater target recognition. To improve the accuracy and speed of recognition, the structure of YOLOv4 is modified by replacing the upsampling module with a deconvolution module and by incorporating depthwise separable convolution into the network. Moreover, the training set used in the YOLO network is preprocessed by using a modified mosaic augmentation, in which the gray world algorithm is used to derive two images when performing mosaic augmentation. The recognition results and the comparison with the other target detectors demonstrate the effectiveness of the proposed YOLOv4 structure and the method of data preprocessing. According to both subjective and objective evaluation, the proposed target recognition strategy can effectively improve the accuracy and speed of underwater target recognition and reduce the requirement of hardware performance as well.

**Keywords:** underwater image; enhancement; YOLO neural network; target recognition; mosaic data augmentation





## 1. Introduction

There are countless oil and gas, mineral, biological, and chemical resources in the vast ocean. With the continuous development of science and technology, we have a better understanding of the ocean. In the process of ocean exploration, there are many underwater tasks, such as target positioning [1], biological recognition [2], archaeology [3], object search [4], environment detection [5], device maintenance [6], etc. In accomplishing these tasks, underwater target recognition plays an important role. Underwater target recognition depends on an underwater vision system, which can be divided into acoustic vision systems and optical vision systems [7]. Based on the acquired acoustic information, one can perform image processing, feature extraction, classification, and recognition [8]. However, there are some defects in underwater acoustic vision images, such as noise, low resolution, and less image information [9]. In recent years, with the development of optical equipment, optical vision system-based underwater target recognition has received increasing attention. From the point of view of methodology, many target recognition algorithms have been put forward and applied in underwater scenarios.

Target recognition, also known as target detection, is a kind of computer vision task. It aims to identify specific types of visual objects (such as human, animal, or vehicle) in digital images. Most of the early target recognition algorithms are based on handcrafted features. Viola and Jones [10] proposed a VJ recognizer, which realized real-time face recognition for the first time. Dalal and Triggs [11] initially proposed the histogram of oriented gradient (HOG) feature descriptor. Felzenszwalb et al. proposed the deformable part model (DPM) as an extension of the HOG recognizer [12] and made various improvements [13]. DPM is regarded as the peak of traditional target recognition. However, as the performance of handcrafted features tends to be saturated, traditional target recognition hits a bottleneck. In recent years, convolutional neural networks (CNNs) have received attention and have been proven as an effective tool for classification [14]. In the application of CNNs to target

recognition, Girshick et al. [15] initially proposed a CNN-based target recognition tool. He et al. [16] developed a spatial pyramid pool network (SPPnet) with a speed more than 20 times that of RCNN (Regions with CNN factures), without sacrificing recognition accuracy. In 2015, Redmon et al. [17] proposed the YOLO (You only look once) network, which is the first one-stage detector in the area of deep learning. Liu et al. [18] proposed the second one-stage detector, SSD (single shot multibox detector). Compared to YOLO, the contribution of SSD is the introduction of multi-reference and multi-resolution recognition technology, which significantly improves the recognition accuracy of the one-stage detector, especially for some small objects.

Many researchers have applied target recognition methods to underwater images. In the past, most people used traditional target recognition methods for underwater target recognition. Xu et al. [19] proposed an inhomogeneous illumination field to reduce the backward scattered background noise in underwater environments. Xu et al. [20] proposed an underwater target feature extraction method based on the singular value of a generalized S-Transform module time–frequency matrix. Ma [21] analyzed and extracted the polarization feature, edge feature, and line feature, which are more suitable for underwater environment target detection, and then used the Itti model to generate a saliency map to detect underwater targets. Wang et al. [22] adjusted the regional saliency by simulating the eye movement to change the position of the attention focus and formed the saliency region to achieve underwater object detection. Oliver et al. [23] studied the influence of different underwater point spread functions on the detection of image features using many different feature detectors, and the influence of these functions on the feature detection ability when they are used for matching and target detection. Zhang et al. [24] extracted new combined invariant moments of underwater images as the recognition features of the system, and used an artificial fish swarm algorithm (AFSA) improved neural network as the underwater target classifier. Yang et al. [25] presented a classification algorithm based on multi-feature fusion for moving target recognition. Dubreuil et al. [26] investigated underwater target detection by combining active polarization imaging and optical correlation-based approaches. Yahya et al. [27] proposed a robust target recognition algorithm using bounding box partition to overcome the problem of failure of recognition due to unmatched targets. Liu et al. [28] proposed a feature matching algorithm based on the Hough transform [29] and geometrical features for target detection in special underwater environments. Li et al. [30] proposed an algorithm for the recognition of small underwater targets based on shape characteristics.

In 2015, researchers began to apply deep learning to underwater target recognition. Li et al. [31] first applied deep CNN to underwater detection and constructed an Image-CLEF dataset, which involves 24,277 fish images belonging to 12 classes. Sun et al. [32] proposed a CNN knowledge transfer framework for underwater object recognition and tackled the problem of extracting discriminative features from relatively low-contrast images. Zhou et al. [33] proposed three data augmentation methods for a Faster R-CNN network. Park and Kang [34] improved the YOLO network and proposed a method to accurately classify objects and count them in sequential video images. Arain et al. [35] presented two novel approaches for improving image-based underwater obstacle detection by combining sparse stereo point clouds with monocular semantic image segmentation. Jia and Liu [36] designed a rapid detection method for marine animals based on MSRCR (multiscale Retinex with color restore) and YOLOv3 to solve the problems of insufficient illumination of the underwater environment and slow detection speed in the detection of marine animals. Qiang et al. [37] proposed a target recognition algorithm based on improved SSD to improve the target detection accuracy and speed in the complex underwater environment. Zang et al. [38] used a region-based fully convolutional network (R-FCN) model to identify marine organisms to realize a submersible tool for video information recognition. Liu [39] solved the imbalance of underwater image datasets through a GAN network. Li et al. [40] employed a transfer learning strategy to train the underwater YOLO network and to alleviate the limitation of fish samples.

Generally, deep learning-based underwater target recognition has achieved some progress in recent years. Nevertheless, the powerful ability of deep learning has not been fully exploited. This paper introduces the latest version of YOLO, i.e., YOLOv4, to underwater target recognition. Moreover, in consideration of the particularity of underwater environments, YOLOv4 is modified to improve the speed and accuracy of target recognition. The original upsampling module is replaced with a deconvolution module. At the same time, the depthwise separable convolution is added to the YOLO network to reduce the calculation burden of network recognition and training. Another contribution of the study is that a new data preprocessing method is proposed to improve the accuracy of target recognition. In this method, the background environment factors are taken into account when extracting target features. Mosaic augmentation is employed and improved. It is noted from the literature that, in the process of underwater target recognition, enhancement or restoration algorithms for underwater images are often incorporated into target recognition to improve the accuracy of recognition. For example, Xie et al. [41] applied the dark channel prior model in underwater target recognition. Zhou et al. [42] adopted an adaptive underwater image enhancement algorithm to suppress noise and to improve edge clarity in recognizing moving underwater objects. Liang et al. [43] proposed an algorithm that combines the improved dark channel and MSR (multiscale Retinex) to improve the accuracy of target recognition. Yang et al. [44] proposed a novel underwater image enhancement method to improve the quality of underwater images through color compensation and correction, gamma correction, and brightness de-blurring. Note that, although some underwater image enhancement algorithms yield positive results, the processing speed is limited, which degrades their real-time performance. In this study, to improve the accuracy and speed of underwater target recognition, an improved YOLOv4 is combined with an improved mosaic augmentation in which an image enhancement algorithm, gray world algorithm, is incorporated.

The rest of the paper is organized as follows. In Section 2, the fundamentals of gray world algorithm, YOLO, mosaic augmentation, and the CIoU loss function are described. In Section 3, the improvement of YOLOv4 and modification of mosaic augmentation are explained. In Section 4, some examples are treated with the proposed method to verify its effectiveness. The final section contains the conclusions.

## 2. Fundamentals

### 2.1. Gray World Algorithm

As is known, underwater images are mostly affected by the refraction and absorption of light. Therefore, an image of objects in water presents a blue-green tone. At the same time, the scattering of light induced by particles in the water makes the image details blurred and the surface atomized. The temperature of water also affects the propagation of light, leading to light scattering [45]. In summary, most underwater images have problems of color deviation, blur, and atomization. As a result, it is difficult to extract the features of underwater images.

To alleviate the color deviation, the gray world algorithm is used in this study. This algorithm is based on the gray world hypothesis [46]. The hypothesis holds that the average reflection of light by natural objects is a fixed value in general, and this fixed value is approximately considered "gray". The gray world algorithm is widely used owing to two advantages. Firstly, for a single-tone image—for example, a greenish underwater image—the enhancement effect of gray world algorithm is satisfactory. The second advantage of the gray world algorithm is the low computational cost [47]. The gray world algorithm can be described as follows.

First, the average values of the three channels are calculated:

$$\begin{cases} R_{avg} = \frac{1}{m}\sum_0^{m-1} R \\ G_{avg} = \frac{1}{m}\sum_0^{m-1} G \\ B_{avg} = \frac{1}{m}\sum_0^{m-1} B \end{cases}, \tag{1}$$

where $R$, $G$, $B$, represent the red, green, and blue pixel values of each pixel; $R_{avg}$, $G_{avg}$, $B_{avg}$ are the average of red, green, and blue channels of all pixels. $m$ is the number of pixels in the image. Based on Equation (1), the mean value of RGB can be obtained as:

$$K = \left( R_{avg} + G_{avg} + B_{avg} \right) / 3,$$ (2)

Then, the gain of each channel relative to $K$ can be determined:

$$\begin{cases} K_R = K/R_{avg} \\ K_G = K/G_{avg} \\ K_B = K/B_{avg} \end{cases},$$ (3)

The pixel values of pixels are adjusted one by one according to the obtained gains:

$$\begin{cases} R_{new} = K_R * R \\ G_{new} = K_G * G \\ B_{new} = K_B * B \end{cases},$$ (4)

where $R_{new}$, $G_{new}$, $B_{new}$ represent the new pixel value of each pixel.

### 2.2. YOLO Neural Network

The You only look once (YOLO) neural network was proposed by Redmon et al. in 2015 [18]. It is the first one-stage target detector in the era of neural networks. In this target detector, to improve the detection speed, the "proposal detection + verification" pattern in the two-stage detector (e.g., RCNN, Faster RCNN) is abandoned; only one neural network is used instead. The YOLO network divides the image into several regions with the same size. Then, the network predicts the kinds and probability of objects in bounding boxes in each region. Later, Redmon made a series of improvements on the basis of YOLO, by proposing v2 and v3 versions [48,49].

YOLOv3 was proposed by Redmon in 2018 [50]. Owing to the fast training speed and detection speed, it is often used in practice. YOLOv3 blends several excellent structures, including the Darknet53 network, anchor, and FPN network. The basic framework of the YOLOv3 neural network is shown in Figure 1.

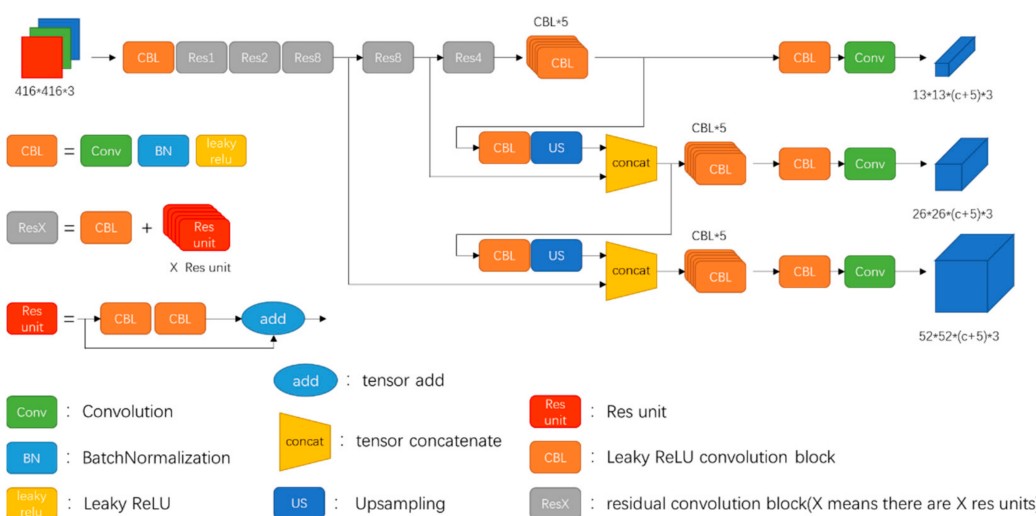

**Figure 1.** Framework of YOLOv3.

YOLOv4 was proposed by Bochkovskiy [50] in 2020 on the basis of YOLOv3. Compared with YOLOv3, YOLOv4 makes many improvements, including changing the backbone network Darknet53 to CSPDarknet53, adding SPP and PANet network, taking CIoU as a part of the loss function, and using mosaic augmentation. The structure of the YOLOv4

neural network is shown in Figure 2. As can be seen, as with YOLOv3, in YOLOv4, the input image is processed as an image with the width and height of 416 × 416. For different sizes of targets, the final output layer of the YOLO network has three sizes of width and height, i.e., 13 × 13, 26 × 26, and 52 × 52. Note that in the output layers, as shown in Figures 1 and 2, for example, 13 × 13 × (c + 5) × 3, *c* represents the number of kinds to be detected; **5** reflects whether there is a target in an anchor box and 4 adjustment parameters of the anchor box; **3** represents three different anchor boxes corresponding to each size of width and height. It can also be seen that the basic convolution blocks of the YOLOv4 network, i.e., CBL and CBM, are similar to YOLOv3. The main difference is that YOLOv4 takes the Mish function as the activation function in the backbone network, while the Leaky ReLU function is used in YOLOv3.

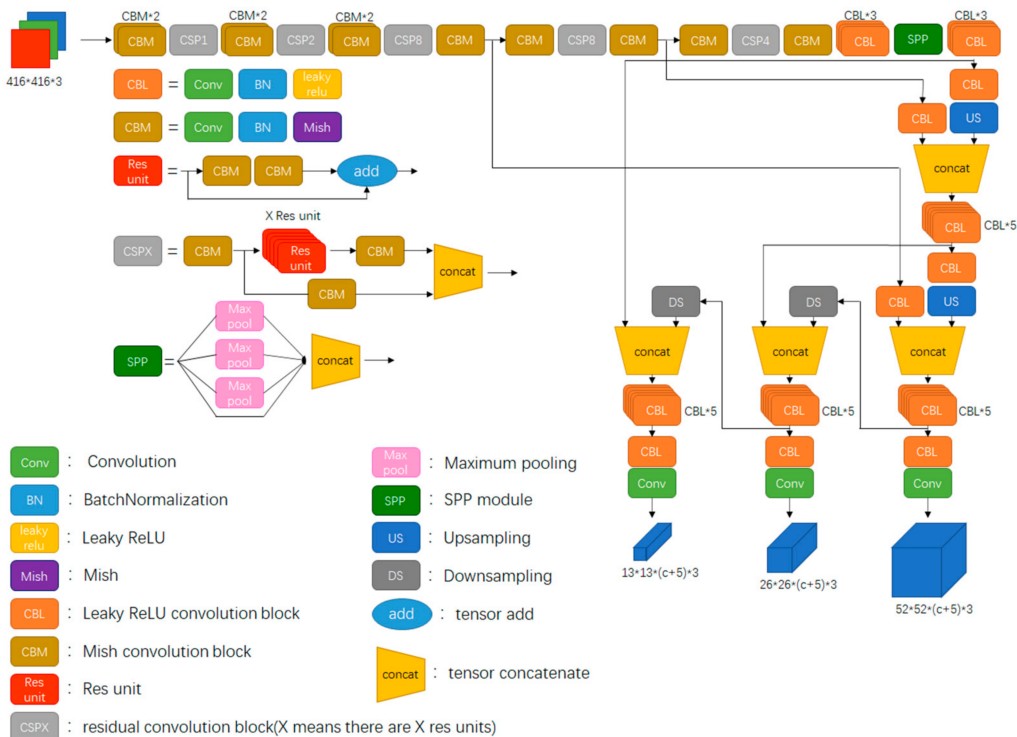

**Figure 2.** Structure of YOLOv4.

A basic convolution block in YOLOv4 is composed of normal convolution, batch normalization, and activation functions.

For the CBL block,

$$x' = \text{Leaky ReLU}(\text{Batch Normalization}(\text{Conv}(x))). \tag{5}$$

For the CBM block,

$$x' = \text{Mish}(\text{Batch Normalization}(\text{Conv}(x))), \tag{6}$$

where $x$ is the input, $x'$ is the output, Conv() is the normal convolution operator, Batch Normalization() is the batch normalization operator, and Leaky ReLU() and Mish() are activation functions.

The Res unit is a basic residual component in both YOLOv3 and YOLOv4. The output feature of the unit is obtained by adding the input feature and the result of two convolutions. In the residual convolution block of YOLOv3, the input feature enters the Res unit after a basic convolution. Then, an output feature is obtained. In YOLOv4, the input feature travels along two routes after a basic convolution. In one route, the result of convolution goes through the Res unit and the basic convolution, while in the other route, the result of

convolution only experiences the basic convolution. The final output feature is obtained by concatenating the results of two routes.

In addition to modifying the backbone network, YOLOv4 also adds the SPP network and modifies the FPN network to the PANet network. The SPP network performs maximum pooling of inputs with different sizes and concatenates the pooling results together to obtain an output. The PANet network can be regarded as an upgrade of the FPN network, in which the structure of downsampling is added.

### 2.3. Mosaic Augmentation

Mosaic augmentation [50] derives from CutMix augmentation [51], which stitches two pictures together into one picture to increase the number of targets and enhance the complexity of the background. It has been proven that such an operation is beneficial to the training of a target detection neural network.

### 2.4. CIoU

IoU (Intersection over Union) is a basic criterion in target recognition, being the ratio of the intersection and union of the predict box and ground truth. In the training of a neural network, sometimes, it is problematic to take IoU as a part of the loss function, especially for the case of IoU = 0 and the case when different intersections have the same IoU value. Therefore, in YOLOv4, CIoU (Complete IoU) [52] is selected to replace IoU as a part of the loss function. The lower the CIoU, the better a predicted box approximates the ground truth. The CIoU part of the loss function and its components are as follows:

$$L_{\text{CIoU}} = 1 - \text{IoU} + \frac{\rho^2(b, b^{\text{gt}})}{c^2} + \alpha\nu, \tag{7}$$

$$\text{IoU} = \frac{|B \cap B^{\text{gt}}|}{|B \cup B^{\text{gt}}|}, \tag{8}$$

$$\nu = \frac{4}{\pi^2}\left(\arctan\frac{w^{\text{gt}}}{h^{\text{gt}}} - \arctan\frac{w}{h}\right)^2, \tag{9}$$

$$\alpha = \frac{\nu}{(1 - \text{IoU}) + \nu}, \tag{10}$$

where $B$, $B^{\text{gt}}$ represent the predict box and the ground truth, respectively; $\rho(b, b^{\text{gt}})$ is the distance between the center point of the predict box and the ground truth; $c$ represents the diagonal distance of the smallest rectangle containing the predict box and the ground truth; $w$, $h$, $w^{\text{gt}}$, $h^{\text{gt}}$ represent the width and height of the predict box and the ground truth, respectively; $\alpha$ is a positive trade-off parameter to measure the consistency of aspect ratio $\nu$. The parameters of CIoU are shown in Figure 3.

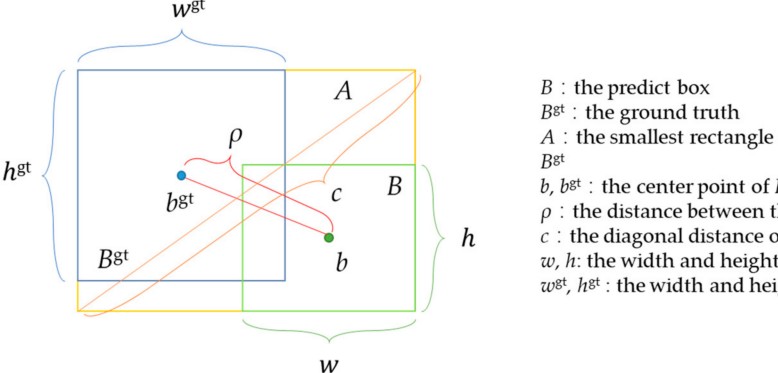

**Figure 3.** Description of CIoU.

## 3. Improvement of YOLOv4

### 3.1. E-Mosaic Augmentation

As is known, there are two main purposes of mosaic augmentation in target recognition. One is to increase the complexity of the image background. The other is to increase the number of objects in an image. Due to the particularity of the underwater environment, the image background is mostly blue-green for underwater images. The complexity of the background is too weak for mosaic augmentation. As a result, the accuracy of target recognition cannot be improved obviously, even if mosaic augmentation is carried out.

In this study, a novel data augmentation method, named after e-mosaic augmentation, is proposed. An image enhancement algorithm, gray world algorithm, is incorporated into mosaic augmentation. Similarly to normal mosaic augmentation, e-mosaic augmentation selects four images. The difference is that, before stitch, the e-mosaic augmentation first enhances two images on the diagonal by using gray world algorithm. The purpose is to increase the complexity of the background, which benefits the training of a target recognition neural network. Figure 4 presents the process of e-mosaic augmentation.

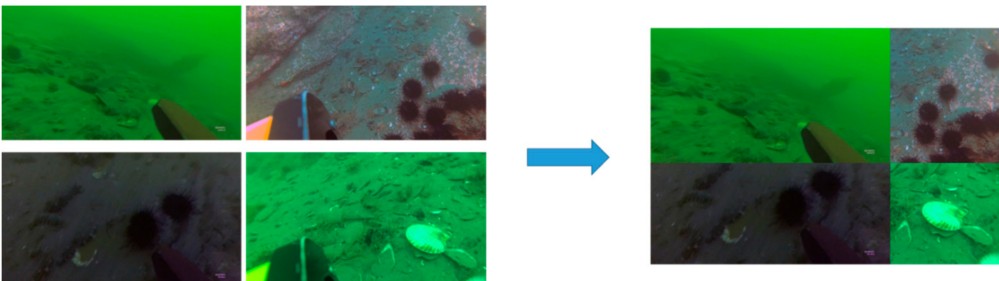

**Figure 4.** E-mosaic augmentation.

### 3.2. YOLOv4-uw

In this study, the structure of YOLOv4 is modified to increase the target recognition accuracy and speed. To improve the accuracy of recognition, the upsampling module in YOLOv4 is replaced with the deconvolution module. To increase the speed of recognition, depthwise separable convolution is used to replace the normal convolution in the Res unit and the normal convolution near the output layer.

Figure 5 shows the difference between upsampling and deconvolution. As can be seen, the upsampling structure can only enlarge the feature size and cannot restore the details. By contrast, the deconvolution structure can not only enlarge the feature size but also restore details. This is because deconvolution is the reverse operation of convolution, which can approximately restore the feature layer to what it was before convolution. Obviously, compared to upsampling, deconvolution is conducive to the restoration of image details.

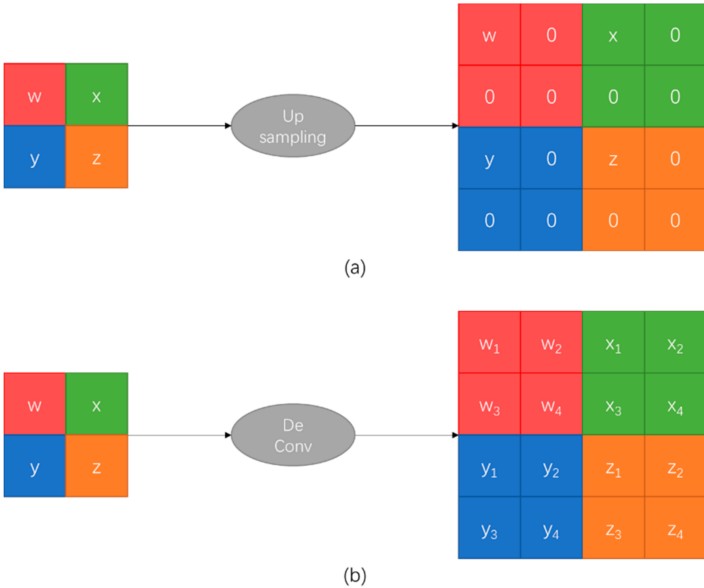

**Figure 5.** Upsampling and deconvolution: (**a**) upsampling; (**b**) deconvolution.

Figure 6 presents the comparison between a normal three-dimensional convolution and a depthwise separable three-dimensional convolution. The left plot is the normal $3 \times 3$ convolution block, which is composed of $3 \times 3$ convolution, batch normalization (BN), and the activation function (AF). The right plot is a $3 \times 3$ depthwise separable convolution block, which is composed of depthwise convolution, point convolution ($1 \times 1$ convolution), batch normalization, and the activation function. Depthwise separable convolution is the core of MobileNet [53]. The difference between depthwise separable convolution and normal three-dimensional convolution is that depthwise separable convolution divides convolution operation into two steps to reduce the amount of convolution calculation. Although the accuracy might be slightly reduced, the speed of training and detection can be usually improved.

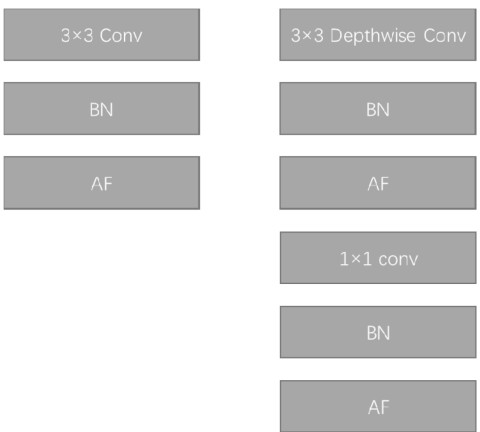

**Figure 6.** A $3 \times 3$ convolution and corresponding depthwise separable convolution.

Figure 7 presents the structure of the modified YOLOv4 in the study, named YOLOv4-uw since this version of YOLOv4 is constructed for the purpose of underwater target recognition. Notably, it is observed that in the used training set and test set, the objects to be recognized are of medium size and small size. To further improve the speed of target recognition, the large-sized output layer in YOLOv4, i.e., with the size of $52 \times 52$, is removed. Correspondingly, the SPP structure is deleted since SPP is introduced in YOLOv4 to deal with various sizes of objects.

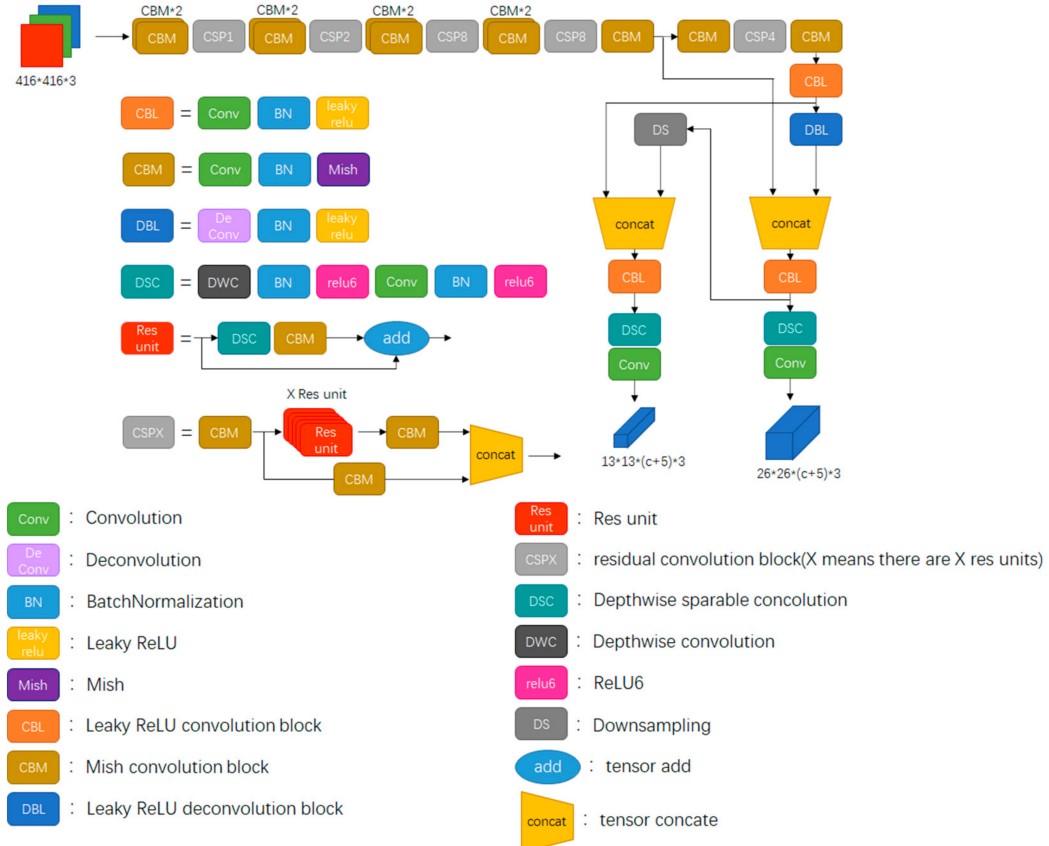

**Figure 7.** Structure of YOLO-uw.

## 4. Results and Discussion

### 4.1. Training and Test Set

The training environment of YOLO is: Windows10; Inter® Core™ i5-10600KF; NVIDIA GeForce RTX 3080; Python3.7; CUDA11.0; Cudnn8.0.4; tensorflow-gpu2.4.0.

The experimental dataset is from the Target Recognition Group of 2020 China Underwater Robot Professional Competition (URPC 2020). In total, it contains 4757 pictures, 3805 of which (80% data) are taken as the training set, while the rest, 952 pictures (20% data), are taken as the test set. The target types are divided into four categories: echinus, starfish, holothurian, and scallop.

### 4.2. Verification of E-Mosaic Augmentation

The proposed e-mosaic augmentation method is verified from objective and subjective aspects. The recognition precision is calculated as the objective evaluation index. The visual detection results are presented to subjectively evaluate the effect of e-mosaic augmentation.

Table 1 shows the precision of recognition results in terms of AP (average precision) and mAP (mean average precision), in which the threshold value of IoU is set to 0.5. The AP value is a comprehensive evaluation metric based on the precision and recall in a category. The precision and recall in a category can be calculated after setting the threshold of IoU. The mAP value is the mean AP of all categories. To verify the effectiveness of e-mosaic augmentation, three different preprocessing methods with respect to the original dataset are considered, including without data preprocessing, data preprocessing by mosaic augmentation, and data preprocessing by proposed e-mosaic augmentation. After the training set is determined, training is performed by using YOLOv4. As shown in Table 1, e-mosaic augmentation can effectively improve the detection accuracy. Compared with the YOLOv4 without any data preprocessing, in the four underwater animal categories, the recognition precision of starfish is improved the most by mosaic and e-mosaic augmentation. Com-

pared with mosaic augmentation, the proposed e-mosaic augmentation leads to a general improvement in terms of both AP and mAP.

**Table 1.** Comparison of precision of recognition results with different preprocessing of training set.

| Preprocessing of Training Set | AP | | | | mAP (%) |
|---|---|---|---|---|---|
| | Scallop | Echinus | Starfish | Holothurian | |
| Without preprocessing | 0.66 | 0.85 | 0.51 | 0.55 | 63.96 |
| mosaic | 0.58 | 0.83 | 0.74 | 0.54 | 67.20 |
| e-mosaic | 0.61 | 0.84 | 0.72 | 0.57 | 68.46 |

Figure 8 presents some recognition results of four underwater animals, in which recognition of scallop is labeled by a blue box; echinus by purple; starfish by yellow; holothurian by red. The plots in the first column of Figure 8a are obtained in the case of the training set without preprocessing. Presented in column Figure 8b are the results after using mosaic augmentation in training. Figure 8c shows the recognition results after using e-mosaic augmentation in training. It can be seen that the recognition results obtained through e-mosaic augmentation-assisted training are the best. Not only are the recognized targets the most, but also the predict box is the most accurate.

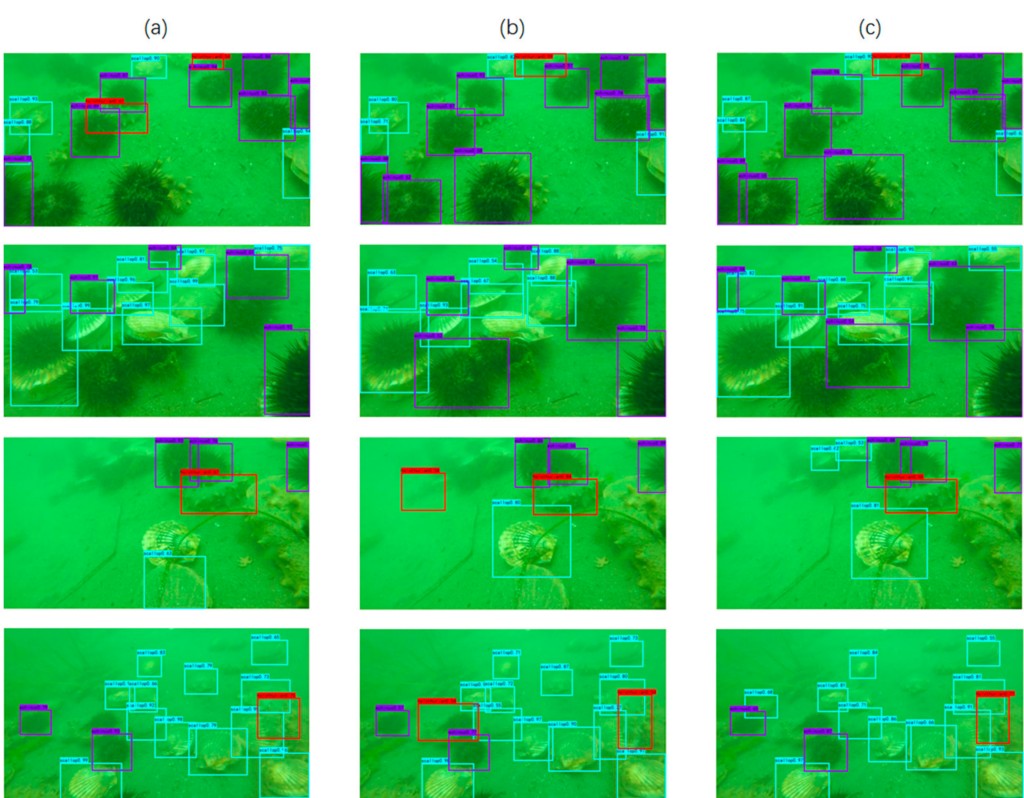

**Figure 8.** Target recognition results: (**a**) without preprocessing training set; (**b**) preprocessing training set by mosaic augmentation; (**c**) preprocessing training set by e-mosaic augmentation.

*4.3. Verification of YOLOv4-uw*

As aforementioned, the large-scale output layer is deleted from the proposed YOLOv4-uw in the study because of the lack of large-scale objects in the available training set. The SPP structure is also removed. Table 2 shows the comparison of YOLOv4-uw with and without the SPP structure. As can be seen from the comparison results, SPP does not improve the detection accuracy. It can be even found that the detection speed in terms of FPS (frame per second) is improved after the SPP structure is deleted.

**Table 2.** Comparison of YOLOv4-uw with and without SPP.

| Preprocessing of Training Set | AP | | | | mAP (%) | Detection Speed (FPS) |
|---|---|---|---|---|---|---|
| | Scallop | Echinus | Starfish | Holothurian | | |
| YOLOv4-uw with SPP | 0.64 | 0.86 | 0.82 | 0.62 | 73.48 | 39 |
| YOLOv4-uw without SPP | 0.64 | 0.86 | 0.83 | 0.68 | 75.34 | 44 |

The proposed YOLOv4-uw neural network is further verified via network performance and hardware requirement. The network performance is reflected by detection accuracy and speed while the hardware requirement refers to model size, parameters, and required calculation ability.

Table 3 shows the comparison between the proposed YOLOv4-uw and other classic networks. The mAP value, time spent in detection, and detection speed are used as evaluation criteria. Notably, when mAP is measured, the IoU threshold is set to 0.5. As can be seen from the comparison results, the target recognition by the YOLOv4-uw network yields a mAP index value of 75.34%, which is nearly 12% higher than YOLOv4. Moreover, YOLOv4-uw outperforms other target recognition networks. In terms of detection time, the proposed YOLOv4-uw takes 2.3 ms, which is 0.5 ms faster than YOLOv4. In terms of FPS, the proposed YOLOv4-uw reaches 44 FPS, which is 9 FPS more than YOLOv4.

**Table 3.** Comparison between YOLOv4-uw and other networks.

| Network | mAP (%) | Time Spent in Detection (ms) | Detection Speed (FPS) |
|---|---|---|---|
| Faster RCNN | 41.98 | 5.7 | 17 |
| SSD | 71.45 | 0.9 | 105 |
| CenterNet | 73.57 | 1.8 | 55 |
| YOLOv3 | 31.52 | 2.1 | 48 |
| YOLOv4 | 63.96 | 2.8 | 35 |
| YOLOv4-uw | 75.34 | 2.3 | 44 |

Table 4 shows the comparison of the required network model size, parameters, and required calculation ability (represented by billion floating point operations per second, Bflop/s) among the proposed YOLOv4-uw and other one-step detectors. The smaller the network is, the lower the requirement of storage space is. As shown in Table 4, the size of YOLOv4-uw is 65 MB, around a quarter of that of YOLOv4. Moreover, the values of the parameters in the YOLOv4-uw are the lowest. BFLOP/s is usually used as a measure of the computing power required by a network. The larger the BFLOP/s is, the higher the requirement of the device is. As can be seen from Table 4, the BFLOP/s of YOLOv4-uw is the smallest by 19.6 Bflop /s, around one third of that of YOLOv4.

**Table 4.** Comparison between YOLOv4-uw and other one-step detectors.

| Network | Model Size (MB) | Total Parameters (M) | Bflop/s |
|---|---|---|---|
| SSD | 92 | 24.2 | 63.2 |
| CenterNet | 125 | 32.7 | 50.5 |
| YOLOv3 | 235 | 61.6 | 65.4 |
| YOLOv4 | 250 | 64.0 | 59.7 |
| YOLOv4-uw | 65 | 16.7 | 19.6 |

*4.4. Verification of the Combination of E-Moasic and YOLOv4-uw*

Table 5 shows the mAP of recognition results by YOLOv4, YOLOv4 with e-mosaic, YOLOv4-uw, and YOLOv4-uw with e-mosaic. Three cases with different IoU threshold values are considered. IoU@0.5 indicates that the IoU threshold is set to 0.5. IoU@0.75 means that the IoU threshold is set to 0.75. IoU@[0.5:0.95] means that when the IoU changes from 0.5 to 0.95, the mAP is tested every 0.05. Then, the average value of 10 measurement

results is taken as the final mAP. As can be seen from the results in Table 5, when the IoU threshold is set to 0.5, the recognition precision by YOLOv4-uw associated with e-mosaic augmentation can reach 76.84%, which is nearly 13% higher than YOLOv4 without e-mosaic augmentation and around 8% higher than YOLOv4 with e-mosaic augmentation. When the IoU threshold is set to [0.5:0.95], it can be seen that under the same data augmentation conditions, the mAP of YOLOv4-uw is around 11% higher than that of YOLOv4. The mAP of the same network (YOLOv4 or YOLOv4-uw) can be increased by around 1.2% by using e-mosaic augmentation.

**Table 5.** Comparison of YOLOv4 and YOLOv4-uw, with and without e-mosaic.

| Condition | mAP | | | |
|---|---|---|---|---|
| | YOLOv4 | | YOLOv4-uw | |
| e-mosaic | No | Yes | No | Yes |
| IoU @0.5 | 63.96 | 68.46 | 75.34 | 76.84 |
| IoU @0.75 | 12.77 | 12.22 | 28.41 | 30.08 |
| IoU @[0.5:0.95] | 25.03 | 26.36 | 36.12 | 37.35 |

## 5. Conclusions

In this paper, an improved YOLOv4 is proposed to improve the accuracy and speed of underwater target recognition. A modified data augmentation is also proposed to improve the accuracy of recognition. Through the comparison results between the proposed YOLOv4-uw and other networks, including YOLOv4, it can be seen that the strategy proposed in this study can effectively improve the recognition accuracy and speed. Moreover, the requirement of hardware performance using the proposed YOLOv4-uw is lower than the others.

It should be noted that only one image enhancement method, i.e., gray world algorithm, is considered in performing e-mosaic augmentation. In the next work, other image enhancement methods will be considered and applied to the preprocessing of the training set. In addition, further optimization of YOLOv4-uw will be conducted to improve the recognition accuracy and speed. Due to the fact that the dataset in this paper consists of medium and small targets, the effectiveness of the neural network proposed here for large targets has not been confirmed. Future work will be concerned with the verification of the proposed YOLO-uw network when dealing with the detection of large-scale underwater objects.

**Author Contributions:** Conceptualization, L.C. and W.L.; methodology, L.C.; software, L.C., M.Z. and S.D.; validation, W.L. and L.Y.; formal analysis, L.C. and S.D.; investigation, L.C. and M.Z.; resources, W.L. and L.Y.; data curation, L.C., M.Z. and S.D.; writing—original draft preparation, L.C.; writing—review and editing, W.L.; visualization, L.C., M.Z. and S.D.; supervision, W.L. and L.Y.; project administration, W.L. and L.Y.; funding acquisition, W.L. and L.Y. All authors have read and agreed to the published version of the manuscript.

**Funding:** This research was funded by the China Fujian Provincial Department of Ocean and Fisheries, Grant MHGX-16; Fujian Provincial Industrial Robot Basic Components Technology Research and Development Center, Grant number 2014H21010011.

**Acknowledgments:** The authors would like to thank the anonymous reviewers for their constructive suggestions, which comprehensively improved the quality of the paper. The authors also would like to thank the support from Fuzhou Ocean Research Institute and the support by Fujian Provincial College Marine Engineering Equipment Design and Manufacturing Engineering Research Center.

**Conflicts of Interest:** The authors declare no conflict of interest.

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
