# Peer review of "Underwater Target Recognition Based on Improved YOLOv4 Neural Network"

_electronics, doi:10.3390/electronics10141634_

Round 1
Reviewer 1 Report
The authors in the paper present a modification for
- A preprocessing method (e-Mosaic): Instead of the regular merge of 4 photos in mosaic, they use gray world to turn 2 diagonal photos to gray scale to increase the complexity of the back ground.
- yolov4-uw: where they modified yolov4 by changing the deconvolution module into an upsampling module and removed the large size output layer.
The authors starts by presenting comprehensive background, and then introducing their work and the results. The work is valuable and could be helpful in detecting underwater targets. However, the following points need to be addressed:
- The paper has some grammatical and linguistic flaws, which makes it hard to follow, the authors are advised to revise that.
- The same information is repeated in several sections, that hinders the flow and is not efficient.
- AP and mAP are not introduced
- Were bigger underwater objects considered in the study or only small sea creatures? is there a limitation to the size of object that could be detected by the yolov4-uw
- It would be good to see a comparison of YOLOv4 and YOLOv4-uw, with and without removing the SSP structure in terms of detection accuracy and speed, with small and bigger objects.
Otherwise, the work is solid and according to the presented data, the yolov4-uw could be helpful in detecting underwater objects.
Author Response
The authors would like to express their sincere appreciation to the reviewer for his/her comments, the effort and time provided in helping us to improve the quality of the paper. The authors revise the paper carefully according to the comments and suggestions raised by the reviewer.
Comments: The authors in the paper present a modification for
- A preprocessing method (e-Mosaic): Instead of the regular merge of 4 photos in mosaic, they use gray world to turn 2 diagonal photos to gray scale to increase the complexity of the back ground.
- yolov4-uw: where they modified yolov4 by changing the deconvolution module into an upsampling module and removed the large size output layer.
The authors starts by presenting comprehensive background, and then introducing their work and the results. The work is valuable and could be helpful in detecting underwater targets. However, the following points need to be addressed.
Response: Thanks for the comments. In the following, we provide specific responses to his/her suggestions, explaining how the paper is revised.
Concern 1: The paper has some grammatical and linguistic flaws, which makes it hard to follow, the authors are advised to revise that.
Response: Thanks for the comment. The authors revise the manuscript carefully by checking English writing. The authors try their best to correct grammatical errors and typos in the paper. To improve its readability, measures are taken including adding or removing some words and sentences, and rewriting some statements. All changes have been highlighted in the revision.
Concern 2: The same information is repeated in several sections, that hinders the flow and is not efficient.
Response: Thanks for pointing out the mistakes. Indeed, some information has been repeated in several sections. In the revision, the authors try their best to make the statements in the paper more concise by correcting the repetition.
- line 163, subsection 2.2, "...and only one neural network is used in target recognition" has been revised as "...only one neural network is used instead".
- line 241, subsection 3.1, "In underwater images, due to the particularity of underwater environment, the image background is mostly blue-green and similar for most underwater images. " has been revised as "Due to the particularity of underwater environment, the image background is mostly blue-green for underwater images."
- line 249, subsection 3.1, "The purpose is to increase the background difference in the processed image and to increase the complexity of the background, " has been revised as "The purpose is to increase the complexity of the background,"
- line 319, subsection 4.2, the statement "Figure 10 presents some recognition results of four underwater animals in underwater images, "has been revised as "Figure 8 presents some recognition results of four underwater animals,"
- in the paragraph before Table 4, subsection 4.3, the repetition "Table 5 also shows the comparison of the computational complexity between the proposed network and other one-step detectors." has been deleted in the revision.
Concern 3: AP and mAP are not introduced
Response: Thanks for the comment. In the revised manuscript, the explanation of AP and mAP has been added, on lines 304-308, page 9.
Concern 4: Were bigger underwater objects considered in the study or only small sea creatures? is there a limitation to the size of object that could be detected by the yolov4-uw
Response: Thanks for the comments. Because the available dataset for training involves only medium and small objects, bigger underwater objects are not considered in the proposed yolov4-uw. Nevertheless, the authors would like to mention that the main contributions of the proposed yolov4-uw refer to the substitution of upsampling module and the introductin of depthwise separable convolution. Due to the lack of bigger objects in training set, the proposed yolov4-uw is further simplifed by reducing output layers and removing SSP module. However, this simplification is not representative. Instead, the generalization ability or feasibility of the proposed yolov4-uw should be tested on the basis of more general training set. In the revsised manuscript, it has been mentioned in the conclusion that future work will concern about the verification of the proposed YOLO-uw network when dealing with the detection of large-scale underwater objects.
Concern 5: It would be good to see a comparison of YOLOv4 and YOLOv4-uw, with and without removing the SSP structure in terms of detection accuracy and speed, with small and bigger objects.
Response: Thanks for the suggestion. In the revised mansucript, the comparison of YOLOv4 and YOLOv4-uw, with and without removing the SSP structure in terms of detection accuracy and speed has been added. The results are presented in Table 1, page 10. Corresponding explanation has also been given. Please note that as mentioned above, due to the lack of bigger objects in the available dataset, the detection didnot concern about bigger objects.
Comments: Otherwise, the work is solid and according to the presented data, the yolov4-uw could be helpful in detecting underwater objects.
Response: Thanks a lot.

Reviewer 2 Report
A nice manuscript, well written, and organized. However, to my best knowledge, there are few points that the authors can address for improving the manuscript.
- In the introduction, the Hough transform should be given a reference for uninitiated readers.
- Mosaic augmentation should be given a reference for uninitiated readers. [pg 110]
- What is the advantage of using the Grey world algorithm? It would be wonderful to add a few lines describing the advantages.
- CIoU section needs a bit more explanation. So many concepts are used there without explanation or at least reference. It will be easier for the students or non-expert readers to follow if explanations/references are given to these concepts.
- I cannot find Table 1 in the manuscript.
- Please explain the Table 2 the terms AP and mAP.
- The result section needs more organization and explanation for a better understanding.
Author Response
The authors would like to express their sincere appreciation to the reviewer for his/her comments, the effort and time provided in helping us to improve the quality of the paper. The authors revise the paper carefully according to the comments and suggestions raised by the reviewer.
Comments: A nice manuscript, well written, and organized. However, to my best knowledge, there are few points that the authors can address for improving the manuscript.
Response: Thanks for the comment.
Concern 1: In the introduction, the Hough transform should be given a reference for uninitiated readers.
Response: Thanks for the suggestion. In the revised manuscript, a reference, Ref. [30], has been added to the Hough transform.
Concern 2: Mosaic augmentation should be given a reference for uninitiated readers.
Response: Thanks for the suggestion. In the revised manuscript, a reference, Ref. [51], has been added to Mosaic augmentation.
Concern 3: What is the advantage of using the Grey world algorithm? It would be wonderful to add a few lines describing the advantages.
Response: Thanks for the comment and suggestion. In the revised manuscript, the remarks on the advantages of Grey world algorithm have been added, in the second paragraph of subsection 2.1.
Concern 4: CIoU section needs a bit more explanation. So many concepts are used there without explanation or at least reference. It will be easier for the students or non-expert readers to follow if explanations/references are given to these concepts.
Response: Thanks for the comment and suggestion. In the revised manuscript, the exaplanation of CIoU has been supplemented, in the first paragraph of subsection 2.4. Moreover, a figure of the description of CIoU has been added, shown as Figure 3. From this figure, the concepts in CIoU can be easily understood.
Concern 5: I cannot find Table 1 in the manuscript.
Response: Thanks for pointing out the mistake in the original manuscript. In editing the original manuscript, Table 1 is deleted for some reason. However, the authors forgot to deleted corresponding statement. In the revised manuscript, Table 1 is Table 2 in the original manuscript.
Concern 6: Please explain the Table 2 the terms AP and mAP.
Response: Thanks for the comment. In the revised manuscript, the explanation of AP and mAP has been added, on lines 305-309, page 9.
Concern 7: The result section needs more organization and explanation for a better understanding.
Response: Thanks for the comment and suggestion. In the revised manuscript, more explanations have been added in the result section, such as the first paragraph of subsection 4.1, the first and second paragraphs of subsection 4.3. For better understanding, some statements are revised, such as the sentences on line 297, lines 309-312, line 327, lines 344-345, lines 364-365. Moreover, some words are revised for better understanding. All changes have been highlighted in the revised manuscript.

Round 2
Reviewer 1 Report
The authors addressed all the concerns, so I don't have further technical concerns. However, there are still several grammatical and linguistic mistakes in the paper that make it hard to follow. Examples are as follow:
- "Then output feature is obtained. " line 203 p 10 an article is missing (an or a the)
- Lots of missing punctuation like dots at the figure names, commas where needed,..etc.The work is solid and helpful just minor changes are needed
Author Response
The authors thank the reviewer very much for his/her help. We revise the manuscript carefully according to the comments and suggestions raised by the reviewer.
General comments: The authors addressed all the concerns, so I don't have further technical concerns. However, there are still several grammatical and linguistic mistakes in the paper that make it hard to follow.
Response: Thanks for the comments. We try out best to further improve the English writing by correcting grammatical and linguistic errors. All changes have been highlighted in the revised manuscript, including
- Line 205, "go through" has been revised as "goes through";
- In many places, commas have been added, in lines 21, 29, 31, 36, 61, 120, 138, 150, 151, 192, 230, 234, 271, 274, 299, 312, 341, and 352.
- Line 365, a semicolon after "set to 0.5" has been replaced by a dot.
Specific comment 1: "Then output feature is obtained. " line 203 p 10 an article is missing (an or a the)
Response: Thanks for pointing out the mistake. In the revised manuscript, the word "an" has been added before "output feature" .
Specific comment 2: Lots of missing punctuation like dots at the figure names, commas where needed,..etc.
Response: Thanks for pointing out the mistake. In the revised manuscript, the missing dots in all figure names have been added. Moreover, commas where needed have also added in corresponding figure names.
